# Global Convergence and Stability of Stochastic Gradient Descent

**Vivak Patel**
Department of Statistics
University of Wisconsin – Madison
Madison, WI 53706
vivak.patel@wisc.edu

**Shushu Zhang**
Department of Statistics
University of Michigan – Ann Arbor
shushuz@umich.edu

**Bowen Tian**
Department of Statistics
The Ohio State University
tian.837@buckeyemail.osu.edu

## Abstract

In machine learning, stochastic gradient descent (SGD) is widely deployed to train models using highly non-convex objectives with equally complex noise models. Unfortunately, SGD theory often makes restrictive assumptions that fail to capture the non-convexity of real problems, and almost entirely ignore the complex noise models that exist in practice. In this work, we demonstrate the restrictiveness of these assumptions using three canonical models in machine learning. Then, we develop novel theory to address this shortcoming in two ways. First, we establish that SGD's iterates will either globally converge to a stationary point or diverge under nearly arbitrary nonconvexity and noise models. Under a slightly more restrictive assumption on the joint behavior of the non-convexity and noise model that generalizes current assumptions in the literature, we show that the objective function cannot diverge, even if the iterates diverge. As a consequence of our results, SGD can be applied to a greater range of stochastic optimization problems with confidence about its global convergence behavior and stability.

## 1 Introduction

Stochastic Gradient Descent (SGD) and its variants are dominant algorithms for solving stochastic optimization problems arising in machine learning, and have expanded their reach to more complex problems from estimating Gaussian Processes [Chen et al., 2020], covariance estimation in stochastic filters [Kim et al., 2021], and systems identification [Hardt et al., 2016, Zhang and Patel, 2020]. Accordingly, understanding the behavior of SGD and its variants has been crucial to their reliable application in machine learning and beyond. As a result, the theory of these methods has greatly advanced, most notably for SGD as it is the basis for, and simplest of, these methods. Indeed, SGD has been analyzed from many perspectives: global convergence analysis [Lei et al., 2019, Gower et al., 2020, Khaled and Richtárik, 2020, Mertikopoulos et al., 2020, Patel, 2021], local convergence analysis [Mertikopoulos et al., 2020], greedy and global complexity analysis [Gower et al., 2020, Khaled and Richtárik, 2020], asymptotic weak convergence [Wang et al., 2021], and saddle point analysis [Fang et al., 2019, Mertikopoulos et al., 2020, Jin et al., 2021].

While all of these perspectives add new dimensions to our understanding of SGD, the global convergence analysis of SGD is the foundation as it dictates whether local analyses, complexity analyses or saddle point analyses are even warranted. As surveyed in Patel [2021], these current global

36th Conference on Neural Information Processing Systems (NeurIPS 2022).

convergence analyses of SGD make a wide variety of assumptions, most commonly: (1) the objective function is bounded from below, (2) the gradient function is globally Lipschitz continuous, (3) the stochastic gradients are unbiased, and (4) the variance of the stochastic gradients are bounded. While the first and third assumption are generally reasonable,[1] the second and fourth assumptions *and their more recent generalizations* are not usually applicable to machine learning problems as we now demonstrate through three simple examples. Note, in the first two examples, we make use of a penalty function, which can be removed without impacting the result.

**Example 1: Feed Forward Neural Network.** Consider the example $(Y, Z)$ where $Y$ is a binary label and $Z$ is a feature vector. We will attempt to predict $Y$ from $Z$ using a simple multi-layer feed forward network as shown in Fig. 1. The next result states that for a simple distribution over the example space and for a simple, archetype network, the gradient function is not globally Lipschitz continuous, nor does it satisfy the (possibly) more general $(L_0, L_1)$-smooth assumption [Zhang et al., 2019, Definition 1, Assumption 3]. Moreover, the variance of the stochastic gradients is unbounded. See Appendix A.2 for a proof.

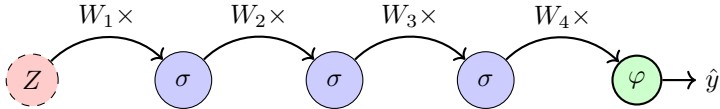

Figure 1: A diagram of a simple feed forward network for binary classification.

**Proposition 1.** *Consider the feed forward network in Fig. 1 with $\sigma$ linear and $\varphi$ sigmoid trained with a binary cross-entropy loss with a Ridge penalty. There exists a finite, discrete distribution for $(Y, Z)$ such that the risk function's gradient is not globally Lipschitz continuous, nor does it satisfy the $(L_0, L_1)$-smooth assumption. Moreover, the variance of the stochastic gradients is not bounded.*

**Example 2: Recurrent Neural Network.** Consider the example $(Y, Z_0, Z_1, Z_2, Z_3)$ where $Y$ is a binary label and $\{Z_0, Z_1, Z_2, Z_3\}$ are sequentially observed. We will attempt to predict $Y$ from $Z$ using a simple recurrent network as shown in Fig. 2. The next result states that for a simple distribution over the example space and for a simple, archetype network, the training function violates the aforementioned assumptions. See Appendix A.3 for a proof.

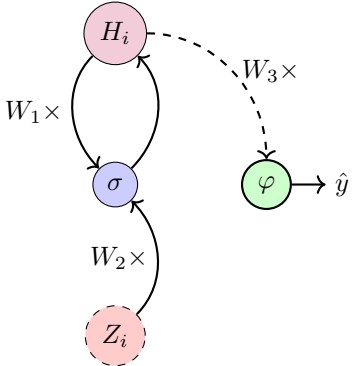

Figure 2: A diagram of a recurrent neural network for a binary classification task.

**Proposition 2.** *Consider the recurrent network in Fig. 2 with $\sigma$ linear and $\varphi$ sigmoid trained with a binary cross-entropy loss with a Ridge penalty. There exists a finite, discrete distribution for $(Y, Z_0, Z_1, Z_2, Z_3)$ such that the risk function's gradient is not globally Lipschitz continuous, nor does it satisfy the $(L_0, L_1)$-smooth assumption. Moreover, the variance of the stochastic gradients is not bounded.*

**Example 3: Poisson Regression.** Consider fitting a Poisson regression model by the standard maximum likelihood framework using independent copies of the example $(Y, Z)$, where $Y$ is a count

---

[1]See Bottou et al. [2018] for a simple relaxation of unbiased stochastic gradients.

response variable and $Z$ is a predictor. The next result states that for a very nice Poisson regression problem, the stochastic gradients violate the bounded variance assumption, its generalization [Bottou et al., 2018, Assumption 4.3c], and, in turn, its generalization, expected smoothness [Khaled and Richtárik, 2020, Assumption 2]. See Appendix A.4 for a proof.

**Proposition 3.** *Let $Y$ and $Z$ be independent Poisson random variables with mean one. Consider estimating a Poisson regression model of $Y$ as a function of $Z$. Then, the gradient function is not globally Lipschitz continuous, nor does it satisfy the $(L_0, L_1)$-smooth assumption. Moreover, the variance of the stochastic gradients is not bounded, does not satisfy [Bottou et al., 2018, Assumption 4.3c], nor does it satisfy [Khaled and Richtárik, 2020, Assumption 2].*

As these three examples show, global convergence analyses that make use of the aforementioned assumptions do not apply to these canonical examples of machine learning problems. In fact, to our knowledge and as summarized in Table 1, *there are no global convergence analyses of SGD* that apply to these examples.

---

**The Problem:**
As a result, we do not have guarantees about how SGD behaves on these simple machine learning problems, which calls into question what SGD and its variants are doing on more complicated machine learning models.

---

**Contributions.** To address this problem,

1. We relax the global Lipschitz continuous gradient assumption, the bounded variance assumption, and their aforementioned generalizations to assumptions that are applicable to the examples above. Specifically, we relax the global Lipschitz continuity assumption and the $(L_0, L_1)$-smooth assumption to local $\alpha$-Hölder continuity of the gradient for $\alpha \in (0, 1]$ (see Assumption 2), which is even a relaxation even for deterministic gradient algorithms (c.f. Nocedal and Wright [2006],Theorems 3.2, 3.8, 4.5, 4.6). For the $\alpha$ in the local Hölder assumption, we also relax the bounded variance assumption to only require that the $(1 + \alpha)$-moment of the stochastic gradient is bounded by an *arbitrary* upper semi-continuous function (see Assumption 4). Our assumption allows stochastic gradients whose noise may not have a variance. Moreover, our assumption generalizes the noise assumption of Bottou et al. [2018], the expected smoothness assumption of Gower et al. [2020] and Khaled and Richtárik [2020], and the noise assumption of Asi and Duchi [2019]. We also point out that we do not require the coercivity or asymptotic flatness assumptions that are commonly considered in the analysis of SGD (e.g., Mertikopoulos et al. [2020], Assumptions 2 and 3).

2. Owing to the relaxation in the assumptions, we cannot apply the standard analysis as the local Hölder constant and the iterate difference are conditionally dependent random variables (see the discussion after Lemma 1). As a result, by generalizing our previous techniques in Patel [2021], Patel and Zhang [2021] to the $\alpha$-Hölder continuous setting, we innovate a new analysis strategy (see Section 4.1) to prove that, with probability one, either SGD's iterates will converge to a stationary point *or* they diverge (see Theorem 2). Importantly, our new analysis strategy can be broadly applied even to deterministic algorithms to relax the assumptions found in the literature.

3. The divergence component of our Theorem 2 is somewhat disconcerting as we cannot say exactly what happens if the iterates diverge. To ameliorate this concern, we add an additional assumption (see Assumption 5) and introduce another analysis strategy (see Section 4.2) to prove, even if the iterates are diverging, the objective function converges to a finite random variable with probability one (see Theorem 3).

Our results *do not* supply a rate of convergence as this is impossible for the broad class of nonconvex functions and the generality of the noise models studied in this work [Wolpert and Macready, 1997]; in other words, we can always construct a nonconvex objective function, a noise process, and choose an initialization such that any rate of convergence statement will be violated. Indeed, we find it remarkable that it is even possible to provide a global convergence statement for such a broad class of nonconvex functions and general noise models.

---

[2]In Majewski et al. [2018], this is implied by their third assumption.

Table 1: A survey of influential, recent global analyses of SGD and their dependence on the two assumptions that are either individually or both violated by the simple examples discussed in Section 1.

| Assumption | Works Depending on the Assumption |
|---|---|
| Global Lipschitz or Hölder Continuity of Gradient | Reddi et al. [2016a], Ma and Klabjan [2017], Zhou et al. [2018], Bassily et al. [2018], Lei et al. [2019], Li and Orabona [2019], Gower et al. [2020], Khaled and Richtárik [2020], Mertikopoulos et al. [2020], Patel [2021], Jin et al. [2021], Wang et al. [2021]. |
| Bounded Variance of Stochastic Gradients | Reddi et al. [2016b], Ma and Klabjan [2017], Majewski et al. [2018], Hu et al. [2019], Bi and Gunn [2019], Zou et al. [2019], Mertikopoulos et al. [2020].[2] |

**Limitations.** We make note of two important limitations in our work. First, we do not consider the important case of nonsmoothness in this work as we require that the gradients of the stochastic optimization function are continuous. However, we note that if the results of Bianchi et al. [2022] are broadly applicable, then SGD never observes a point of nonsmoothness and our results would then be applicable. Second, we do not have a simple interpretation of Assumption 5—though it seems to have a close relative in another analysis (see Wang and Wu [2020])—, nor have we been able to construct a relevant counterexample that can illuminate the limitations of this assumption.

## 2 Stochastic Optimization

We consider solving the optimization problem

$$\min_{\theta \in \mathbb{R}^p} \{ F(\theta) := \mathbb{E}\left[ f(\theta, X) \right] \}, \tag{1}$$

where $F$ maps $\mathbb{R}^p$ into $\mathbb{R}$; $f$ maps $\mathbb{R}^p$ and the co-domain of the random variable $X$ into $\mathbb{R}$; and $\mathbb{E}$ is the expectation operator. As we require gradients, we take $F$ and $f$ to differentiable in $\theta$, and denote its derivatives with respect to $\theta$ by $\dot{F}(\theta)$ and $\dot{f}(\theta, X)$. With this notation, we make the following general assumptions about the deterministic portion of the objective function.

**Assumption 1.** There exists $F_{l.b.} \in \mathbb{R}$ such that $\forall \theta \in \mathbb{R}^p$, $F_{l.b.} \leq F(\theta)$.

**Assumption 2.** There exists $\alpha \in (0, 1]$ such that $\dot{F}(\theta)$ is locally $\alpha$-Hölder continuous.

*Remark* 1. For $\dot{F}$ to be locally $\alpha$-Hölder continuous for some $\alpha \in (0, 1]$, for every compact set $K \subset \mathbb{R}^p$ there exists a constant $L \geq 0$ such that for every $\theta, \psi \in K$,

$$\left\| \dot{F}(\theta) - \dot{F}(\psi) \right\|_2 \leq L \left\| \theta - \psi \right\|_2^\alpha. \tag{2}$$

*Remark* 2. As an example, an empirical risk minimization problem for a deep neural network with twice continuously differentiable activation functions with a twice continuously differentiable loss function will readily satisfy the above conditions.

Assumptions 1 and 2 would even be considered mild in the context of non-convex deterministic optimization, in which it is also common to assume that the objective function has well-behaved level sets [e.g., Nocedal and Wright, 2006, Theorems 3.2, 3.8, 4.5, 4.6]. Importantly, Assumption 2 relaxes the common restrictive assumption of globally Hölder continuous gradient functions that is common in other analyses (see Table 1).

Our final step is to make some assumptions about the stochastic portion of the objective function. The first assumption requires the stochastic gradients to be unbiased, which can readily be relaxed [Bottou et al., 2018]. The second assumption allows for a generic noise model for an $\alpha$-Hölder continuous gradient function, and even allows for the second moment to not exist when $\alpha < 1$ [c.f. Wang et al., 2021, which requires a decomposition of the noise term that we do not require].

**Assumption 3.** For all $\theta \in \mathbb{R}^p$, $\dot{F}(\theta) = \mathbb{E}[\dot{f}(\theta, X)]$.

**Assumption 4.** Let $\alpha \in (0, 1]$ be as in Assumption 2. There exists an upper semi-continuous function $G(\theta)$ such that $\mathbb{E}[\| \dot{f}(\theta, X) \|_2^{1+\alpha}] \leq G(\theta)$.

*Remark* 3. For $G(\theta)$ to be upper semi-continuous, then for all $g > 0$, $\{\theta \in \mathbb{R}^p : G(\theta) < g\}$ are open in $\mathbb{R}^p$.

---

We will show that Assumptions 1 to 4 are sufficient for a global convergence result (see Theorem 2).

---

*Remark* 4. As shown in §A, our examples from §1 satisfy Assumptions 1 to 4.

*Remark* 5. It is entirely possible that $\mathbb{E}[\|\dot{f}(\theta, X)\|_2^{1+\alpha}]$ is (at least) upper semi-continuous, and to set $G(\theta)$ equal to this function. In the case that $\mathbb{E}[\|\dot{f}(\theta, X)\|_2^{1+\alpha}]$ is not upper semi-continuous, it is possible to specify $G(\theta)$ as the upper envelope of $\mathbb{E}[\|\dot{f}(\theta, X)\|_2^{1+\alpha}]$ (i.e., its limit supremum function). However, it is unlikely that $\mathbb{E}[\|\dot{f}(\theta, X)\|_2^{1+\alpha}]$ nor its upper envelope are easy to specify explicitly, and it is more likely to be able to find an upper bound.

*Remark* 6. We use Assumption 4 to conclude that on any compact set, the $1 + \alpha$ moment of the stochastic gradient is bounded. Of course, we can assume this directly (i.e., on any compact set, the $1 + \alpha$ moment is bounded), which, at first glance, appears to be a relaxation. However, if we assume that on any compact set, the $1 + \alpha$ moment is bounded, we can use this to construct a $G(\theta)$ that is upper semi-continuous. Thus, the two assumptions are equivalent.

*Remark* 7. A simple example that shows the utility of Assumption 4 is to optimize $\mathbb{E}[\theta^X]$ where $X$ is an exponential random variable with parameter 1 and $\theta \in [1, u]$ where $u < \exp(1)$. First, it is easy to confirm that the objective function is differentiable and its derivative is globally Lipschitz continuous. Moreover, given that we are on a bounded interval, we conclude that the derivative is globally $\alpha$-Hölder continuous for any $\alpha \in (0, 1]$; therefore, we are free to choose the $\alpha$ as we see fit. Now, when $u < \exp(1/2)$, we have that second moment of the stochastic gradient function exists. However, when $\exp(1/2) < u < \exp(1)$, only smaller moments of the stochastic gradient will exist. Specifically, only for $u < \exp(1/(1+\epsilon))$ with $\epsilon \in (0, 1)$ will the $1 + \epsilon$ moment of the stochastic gradient will exist. Thus, depending on the size of our interval, we may not have the existence of the second moment, and, consequently, we may not have the existence of the variance.

In order to show that the objective function cannot diverge (i.e., to prove stability), we will need an additional assumption. This assumption will relate the gradient function, noise model and variation on the local Hölder constant. To begin, we define the variation on the local Hölder constant. Let $\alpha \in (0, 1]$ be as in Assumption 2 and $\epsilon > 0$ be arbitrary, and define

$$
\mathcal{L}_\epsilon(\theta) = \begin{cases} \sup_\varphi \left\{ \dfrac{\|\dot{F}(\varphi) - \dot{F}(\theta)\|_2}{\|\varphi - \theta\|_2^\alpha} : \|\varphi - \theta\|_2 \leq (G(\theta) \vee \epsilon)^{\frac{1}{1+\alpha}} \right\} & \text{if this quantity is nonzero} \\ \epsilon & \text{otherwise,} \end{cases} \tag{3}
$$

where $\vee$ indicates the maximum between two quantities. Note, the choice of $\epsilon$ is irrelevant, and they can be distinct for the two cases in the definition of $\mathcal{L}_\epsilon$, but we fix them to be the same for simplicity. Importantly, the quantity, $\mathcal{L}_\epsilon$, is defined at every parameter $\theta$ under Assumption 2.

With this quantity, we can state a nonintuitive, technical assumption that is needed to prove stability.

**Assumption 5.** There exists $C_1, C_2, C_3 \geq 0$ such that, $\forall \theta \in \mathbb{R}^p$,

$$
\mathcal{L}_\epsilon(\theta) G(\theta) + \alpha \left( \frac{\left\| \dot{F}(\theta) \right\|_2^{1+\alpha}}{\mathcal{L}_\epsilon(\theta)} \right)^{1/\alpha} \leq C_1 + C_2(F(\theta) - F_{l.b.}) + C_3 \left\| \dot{F}(\theta) \right\|_2^2. \tag{4}
$$

Assumption 5 generalizes Assumption 4.3(c) of Bottou et al. [2018], which is satisfied for a large swath of statistical models. Moreover, Assumption 5 generalizes the notion of expected smoothness [see Khaled and Richtárik, 2020, for a history of the assumption], which expanded the optimization problems covered by the theory of Bottou et al. [2018]. Note, Assumption 5 is about the asymptotic properties of the stochastic optimization problem as the left hand side of the inequality in Assumption 5 can be bounded inside of any compact set. Thus, Assumption 5 covers a variety of asymptotic behaviors, such as $\exp(\|\theta\|_2^2), \exp(\|\theta\|_2), \|\theta\|_2^r$ for $r \in \mathbb{R}, \log(\|\theta\|_2 + 1)$, and $\log(\log(\|\theta\|_2 + 1) + 1)$. Therefore, Assumption 5 holds for functions with a variety of different asymptotic behaviors.

---

We will show that Assumptions 1 to 5 are sufficient for a stability result (see Theorem 3).

---

Now that we have specified the nature of the stochastic optimization problem, we turn our attention to the algorithm used to solve the problem, namely, stochastic gradient descent (SGD).

# 3    Stochastic Gradient Descent

SGD starts with an arbitrary initial value, $\theta_0 \in \mathbb{R}^p$, and generates a sequence of iterates $\{\theta_k : k \in \mathbb{N}\}$ according to the rule

$$\theta_{k+1} = \theta_k - M_k \dot{f}(\theta_k, X_{k+1}), \tag{5}$$

where $\{M_k : k + 1 \in \mathbb{N}\} \subset \mathbb{R}^{p \times p}$; and $\{X_k : k \in \mathbb{N}\}$ are independent and identically distributed copies of $X$. Importantly, $\{M_k\}$ cannot be arbitrary, and the following properties specify a generalization of the Robbins and Monro [1951] conditions for matrix-valued learning rates [c.f. Patel, 2021].

The first condition requires a positive learning rate, and imposes symmetry to ensure the existence of real eigenvalues.

**Property 1.** $\{M_k : k + 1 \in \mathbb{N}\}$ are symmetric, positive definite matrices.

The next two properties are a natural generalization of the Robbins-Monro conditions. Let $\alpha \in (0, 1]$ be as in Assumption 2.

**Property 2.** Let $\lambda_{\max}(\cdot)$ denote the largest eigenvalue of a symmetric, positive definite matrix. Then, $\sum_{k=0}^{\infty} \lambda_{\max}(M_k)^{1+\alpha} =: S < \infty$.

**Property 3.** Let $\lambda_{\min}(\cdot)$ denote the smallest eigenvalue of a symmetric, positive definite matrix. Then, $\sum_{k=0}^{\infty} \lambda_{\min}(M_k) = \infty$.

> We will show that Properties 1 to 3 are sufficient for a global convergence result (see Theorem 2).

The final property ensures the stability of the condition number of $\{M_k\}$. Note, this property is readily satisfied for scalar learning rates satisfying the Robbins-Monro conditions.

**Property 4.** Let $\kappa(\cdot)$ denote the ratio of the largest and smallest eigenvalues of a symmetric, positive definite matrix. Then, $\lim_{k \to \infty} \lambda_{\max}(M_k)^{\alpha} \kappa(M_k) = 0$.

> We will show that Properties 1 to 4 are sufficient for stability (see Theorem 3).

# 4    Global Convergence & Stability

With the stochastic optimization problem and with stochastic gradient descent (SGD) specified, we now turn our attention to what happens when SGD is applied to a stochastic optimization problem. The key step in the analysis of SGD on any objective function is to establish a bound between the optimality gap at $\theta_{k+1}$ with that of $\theta_k$. This step is achieved by using the local Hölder continuity of the gradient function and the fundamental theorem of calculus. Using Assumption 2, we first specify the local Hölder constant.

**Definition 1.** For any $\theta, \varphi \in \mathbb{R}^p$, define

$$L(\theta, \varphi) = \sup_{\psi} \left\{ \frac{\left\| \dot{F}(\psi) - \dot{F}(\theta) \right\|_2}{\|\psi - \theta\|_2^{\alpha}} : \psi \in \overline{B(\theta, \|\varphi - \theta\|_2)} \right\}, \tag{6}$$

where $B(\theta, r)$ is an open ball around $\theta$ of radius $r > 0$, and $\overline{B(\theta, r)}$ is its closure. Moreover, for any $R \geq 0$, let $L_R$ be the supremum of $L(\theta, \varphi)$ for any distinct $\theta, \varphi \in \overline{B(0, R)}$.

*Remark* 8. Note, when the gradient is locally Hölder continuous, $L_R$ is finite for any $R \geq 0$.

With this definition, we can now relate the optimality gap of $\theta_{k+1}$ with that of $\theta_k$ by using the following result and proved in Appendix B.

**Lemma 1.** *Suppose Assumptions 1 and 2 hold. Then, for any $\theta, \varphi \in \mathbb{R}^p$,*

$$F(\varphi) - F_{l.b.} \leq F(\theta) - F_{l.b.} + \dot{F}(\theta)'(\varphi - \theta) + \frac{L(\theta, \varphi)}{1 + \alpha} \|\varphi - \theta\|_2^{1+\alpha}. \tag{7}$$

Now, if we simply set $\varphi = \theta_{k+1}$ and $\theta = \theta_k$ in Lemma 1 and try to take expectations to manage the randomness of the stochastic gradient, we will run into the problem that $L(\theta_k, \theta_{k+1})$ and $\|\theta_{k+1} - \theta_k\|_2$ are potentially dependent,[3] and we cannot compute its expectation. In previous work, this technical challenge is waived away by using a global Hölder constant to upper bound $L(\theta_k, \theta_{k+1})$, which is unrealistic even for simple problems (see Section 1).

To address this technical challenge, we innovate two new strategies for handling the dependence between $L(\theta_k, \theta_{k+1})$ and $\|\theta_{k+1} - \theta_k\|_2$. In both strategies, we follow the same general approach:

1. We begin by restricting our analysis to specific events, which will allow us to decouple $L(\theta_k, \theta_{k+1})$ and $\|\theta_{k+1} - \theta_k\|_2$.
2. With these two quantities decoupled, we will develop a recurrence relationship between the optimality gap at $\theta_{k+1}$ and that of $\theta_k$.
3. We apply this recurrence relationship with refinements of standard arguments or new ones to derive the desired property about the objective function.
4. Finally, we state the generality of the specific events on which we have studied SGD's iterates.

Thus, it follows, we will define two distinct series of events for the two strategies. The first strategy, which we refer to as the pseudo-global strategy, will provide the global convergence analysis. The second strategy, which we refer to as the local strategy, will provide the stability result.

### 4.1 Pseudo-Global Strategy and Global Convergence Analysis

For the first strategy, which supplies the global convergence result, we study SGD on the events

$$\mathcal{B}_k(R) := \bigcap_{j=0}^{k} \left\{ \|\theta_j\|_2 \leq R \right\}, \; k + 1 \in \mathbb{N}, \tag{8}$$

for every $R \geq 0$. We now try to control the optimality gap at iteration $k + 1$ with that of iteration $k$, which will result in two cases.

1. (Case 1) $\mathcal{B}_{k+1}(R)$ holds. We can bound $L(\theta_k, \theta_{k+1})$ by $L_R$, and $G(\theta)$ is also bounded in the ball of radius $R$ about the origin (which follows from $G$ being upper semi-continuous in Assumption 4). As a result, we could then proceed with the analysis in a manner that is similar to having a global Hölder constant.
2. (Case 2) $\|\theta_{k+1}\|_2 > R$ and $\mathcal{B}_k(R)$ holds. In this case, controlling $L(\theta_k, \theta_{k+1})$ is very challenging and, to our knowledge, was not solved before our work.

Our approach for controlling the optimality gap in both cases is supplied in the next lemma, whose proof is in Appendix C.

**Lemma 2.** *Let $\{M_k\}$ satisfy Property 1. Suppose Assumptions 1 to 4 hold. Let $\{\theta_k\}$ satisfy (5). Then, $\forall R \geq 0$,*

$$\mathbb{E}\left[ [F(\theta_{k+1}) - F_{l.b.}]\boldsymbol{1}\left[\mathcal{B}_{k+1}(R)\right] | \mathcal{F}_k \right] \leq [F(\theta_k) - F_{l.b.}]\boldsymbol{1}\left[\mathcal{B}_k(R)\right]$$
$$- \lambda_{\min}(M_k) \left\|\dot{F}(\theta_k)\right\|_2^2 \boldsymbol{1}\left[\mathcal{B}_k(R)\right] + \frac{L_{R+1} + \partial F_R}{1 + \alpha} \lambda_{\max}(M_k)^{1+\alpha} G_R, \tag{9}$$

*where $G_R = \sup_{\theta \in \overline{B(0,R)}} G(\theta) < \infty$ with $G(\theta)$; and $\partial F_R = \sup_{\theta \in \overline{B(0,R)}} \|\dot{F}(\theta)\|_2 (1 + \alpha) < \infty$.*

With this recursion and standard martingale results [Robbins and Siegmund, 1971, Neveu and Speed, 1975, Exercise II.4], the limit of $[F(\theta_k) - F_{l.b.}]\boldsymbol{1}\left[\mathcal{B}_k(R)\right]$ exists with probability one and is finite for

---

[3]While it is possible that these two terms are independent, we would require a lot more information to determine this and it would likely be on an iterate-by-iterate basis for the general class of problems considered in this work. Thus, in this general setting, we cannot assume independence and need to default to treating these terms as dependent.

every $R \geq 0$. As a result, the limit of $F(\theta_k) - F_{l.b.}$ exists and is finite on the event $\{\sup_k \|\theta_k\|_2 < \infty\}$ (see Corollary 1).

We can also use Lemma 2 to make a statement about the gradient. Specifically, we can show that the limit infimum of $\mathbb{E}[\|\dot{F}(\theta_k)\|_2^2 \mathbf{1}[\mathcal{B}_k(R)]]$ must be zero, which is now a standard argument that mimics Zoutendijk's theorem [Nocedal and Wright, 2006, Theorem 3.2]. By Markov's inequality, this result implies that $\|\dot{F}(\theta_k)\|_2 \mathbf{1}[\mathcal{B}_k(R)]$ gets arbitrarily close to 0 infinitely often (see Lemma 9). To show convergence to zero, however, is not standard. Several strategies have been developed, namely those of Li and Orabona [2019], Lei et al. [2019], Mertikopoulos et al. [2020], Patel [2021], Patel and Zhang [2021]. Unfortunately, the approaches of Li and Orabona [2019], Lei et al. [2019] rely intimately on the existence of a global Hölder constant, while that of Mertikopoulos et al. [2020] requires even more restrictive assumptions. Fortunately, the approach of Patel [2021], Patel and Zhang [2021] can be improved and generalized to the current context (see Lemma 10). Thus, we show that $\lim_{k \to \infty} \|\dot{F}(\theta_k)\|_2 = 0$ on $\{\sup_k \|\theta_k\|_2 < \infty\}$ (see Corollary 2).

Our final step is to clarify the role of $\{\sup_k \|\theta_k\|_2 < \infty\}$ in the asymptotics of SGD's iterates. At first glance, this event seems to imply that the iterates converge to a point. However, owing to the general nature of the noise, it is also possible, say, that the iterates approach a limit cycle or oscillate between points with the same norm. Even beyond this event, the generality of the noise model may allow for substantial excursions between $F_{l.b.}$ and infinity (c.f., a simple random walk, which has a limit supremum of infinity and a limit infimum of negative infinity). Thankfully, we can prove that either the iterates converge to a point or they must diverge—a result that we refer to as the Capture Theorem (see Appendix C).

**Theorem 1** (Capture Theorem). *Let $\{\theta_k\}$ be defined as in (5), and let $\{M_k\}$ satisfy Properties 1 and 2. If Assumption 4 holds, then either $\{\lim_{k \to \infty} \theta_k \text{ exists}\}$ or $\{\liminf_{k \to \infty} \|\theta_k\|_2 = \infty\}$ must occur.*

By putting together the above arguments and results, we can conclude that either SGD's iterates diverge or SGD's iterates converge to a stationary point.

**Theorem 2** (Global Convergence). *Let $\theta_0$ be arbitrary, and let $\{\theta_k : k \in \mathbb{N}\}$ be defined according to (5) with $\{M_k : k+1\}$ satisfying Properties 1 to 3. Suppose Assumptions 1 to 4 hold. Let $\mathcal{A}_1 = \{\liminf_{k \to \infty} \|\theta_k\|_2 = \infty\}$ and $\mathcal{A}_2 = \{\lim_{k \to \infty} \theta_k \text{ exists}\}$. Then, the following statements hold.*

1. *$\mathbb{P}[\mathcal{A}_1] + \mathbb{P}[\mathcal{A}_2] = 1$.*
2. *On $\mathcal{A}_2$, there exists a finite random variable, $F_{\lim}$, such that $\lim_{k \to \infty} F(\theta_k) = F_{\lim}$ and $\lim_{k \to \infty} \dot{F}(\theta_k) = 0$ with probability one.*

*Proof.* By Theorem 1, we have that $\mathbb{P}[\mathcal{A}_1] + \mathbb{P}[\mathcal{A}_2] = 1$. Then, on $\mathcal{A}_2$, Corollaries 1 and 2 imply that $F(\theta_k) \to F_{\lim}$, which is finite, and $\dot{F}(\theta_k) \to 0$. $\qquad \square$

We pause to stress to a fact about Theorem 2: it is nonobvious. To be specific, under such general nonconvexity and noise, we should anticipate any number of asymptotic behaviors for the iterates: convergence to a stationary point, convergence to a nonstationary point, being trapped in a cycle, convergence to a limit cycle, and divergence to infinity. However and very surprisingly, we are able to show that only two possible outcomes can occur: convergence to a stationary point or divergence. Indeed, in previous results [e.g., Mertikopoulos et al., 2020, Patel, 2021], only less specific determinations could be made under much more limited settings.

## 4.2 Local Strategy and Stability Analysis

While Theorem 2 provides a complete global convergence result, it allows for the possibility of diverging iterates. The possibility of divergent iterates raises the spectre of whether the objective function can also diverge along this sequence. That is, there is a possibility that SGD may be unstable, which would be highly unexpected and undesirable, especially when the objective function is coercive (e.g., has an $\ell^1$ penalty on the parameters). To formalize this concept, we define a relevant notion of stability.

**Definition 2.** Stochastic Gradient Descent is stable if

$$\mathbb{P}\left[\limsup_{k\to\infty} F(\theta_k) = \infty\right] = 0, \tag{10}$$

where $\{\theta_k\}$ satisfy (5).

We now state the stopping times that we will use to decouple the relationship between $L(\theta_k, \theta_{k+1})$ and $\|\theta_{k+1} - \theta_k\|_2$. For every $j + 1 \in \mathbb{N}$, define

$$\tau_j = \min\left\{ k : \begin{array}{l} F(\theta_{k+1}) - F_{l.b.} > F(\theta_k) - F_{l.b.} + \dot{F}(\theta_k)'(\theta_{k+1} - \theta_k) \\ \quad + \dfrac{\mathcal{L}_\epsilon(\theta_k)}{1 + \alpha} \|\theta_{k+1} - \theta_k\|_2^{1+\alpha}, \text{ and } k \geq j \end{array} \right\}. \tag{11}$$

Now, we will use (11) to establish the stability of the objective function. Just as we did with $\mathcal{B}_k(R)$, we will derive a recursion on the optimality gap over the events $\{\{\tau_j > k\} : k + 1 \in \mathbb{N}\}$. Of course, just as before, the main challenge in deriving a recursive formula is to address $\{\tau_j = k\}$. Our solution is supplied in the following lemma, whose proof is in Appendix D.

**Lemma 3.** *Let $\{M_k\}$ satisfy Property 1. Suppose Assumptions 1 to 4 hold. Let $\{\theta_k\}$ satisfy (5). Then, for any $j + 1 \in \mathbb{N}$ and $k > j$,*

$$\mathbb{E}\left[ (F(\theta_{k+1}) - F_{l.b.}) \mathbf{1}[\tau_j > k] | \mathcal{F}_k \right] \leq \left( F(\theta_k) - F_{l.b.} - \dot{F}(\theta_k)' M_k \dot{F}(\theta_k) \right) \mathbf{1}[\tau_j > k - 1]$$

$$+ \frac{\lambda_{\max}(M_k)^{1+\alpha}}{1 + \alpha} \left[ \mathcal{L}_\epsilon(\theta_k) G(\theta_k) + \alpha \left[ \frac{\left\| \dot{F}(\theta_k) \right\|_2^{1+\alpha}}{\mathcal{L}_\epsilon(\theta_k)} \right]^{1/\alpha} \right] \mathbf{1}[\tau_j > k - 1]. \tag{12}$$

From Lemma 3, there is a clear motivation for Assumption 5. Indeed, if we apply Assumption 5, Lemma 3 produces the following simple recursive relationship.

**Lemma 4.** *If Assumptions 1 to 5, and Properties 1 and 4 hold, and $\{\theta_k\}$ satisfy (5), then there exists a $K \in \mathbb{N}$ such that for any $j + 1 \in \mathbb{N}$ and any $k \geq \min\{K, j + 1\}$,*

$$\mathbb{E}\left[ (F(\theta_{k+1}) - F_{l.b.})\mathbf{1}[\tau_j > k] | \mathcal{F}_k \right]$$

$$\leq \left( 1 + \lambda_{\max}(M_k)^{1+\alpha} \frac{C_2}{1 + \alpha} \right) (F(\theta_k) - F_{l.b.})\mathbf{1}[\tau_j > k - 1] \tag{13}$$

$$- \frac{1}{2} \lambda_{\min}(M_k) \left\| \dot{F}(\theta_k) \right\|_2^2 \mathbf{1}[\tau_j > k - 1] + \lambda_{\max}(M_k)^{1+\alpha} \frac{C_1}{1 + \alpha}.$$

Just as in the pseudo-global strategy, Lemma 4 can be combined with standard martingale results [Robbins and Siegmund, 1971, Neveu and Speed, 1975, Exercise II.4] to conclude that the limit of $F(\theta_k)$ exists and is finite on the event $\cup_{j=0}^\infty \{\tau_j = \infty\}$ (see Corollary 3). Also as in the pseudo-global strategy, by improving on the arguments in Patel [2021], Patel and Zhang [2021], we show that $\liminf_k \dot{F}(\theta_k) = 0$ on the event $\cup_{j=0}^\infty \{\tau_j = \infty\}$ (see Lemma 14).

Finally, we show that $\cup_{j=0}^\infty \{\tau_j = \infty\}$ is a probability one event (see Theorem 5). This statement should not come as a surprise on the event $\{\lim_k \theta_k \text{ exists}\}$, but is slightly surprising that it must also hold on $\{\lim_k \|\theta_k\|_2 = \infty\}$. By combining these results, we can conclude as follows.

**Theorem 3** (Stability). *Let $\theta_0$ be arbitrary, and let $\{\theta_k : k \in \mathbb{N}\}$ be defined according to (5) with $\{M_k : k + 1\}$ satisfying Properties 1 to 4. Suppose Assumptions 1 to 5 hold. Then,*

1. *There exists a finite random variable, $F_{\lim}$, such that $\lim_{k\to\infty} F(\theta_k) = F_{\lim}$ with probability one;*
2. *$\liminf_{k\to\infty} \dot{F}(\theta_k) = 0$ with probability one.*

*Proof.* Using Corollary 3, we conclude that $\exists F_{\lim}$ that is finite such that $\lim_k F(\theta_k) = F_{\lim}$ on $\cup_{j=0}^\infty \{\tau_j = \infty\}$. Using Lemma 14, we conclude that $\liminf_k \dot{F}(\theta_k) = 0$ on $\cup_{j=0}^\infty \{\tau_j = \infty\}$. Finally, we apply Theorem 5 to conclude that $\mathbb{P}[\cup_{j=0}^\infty \{\tau_j = \infty\}] = 1$. $\qquad\square$

We would like to demonstrate a simple example of how we would use Theorem 3. Consider applying SGD to linear regression as specified in Appendix A.1. For this example, it is straightforward to verify that the assumptions of Theorem 3 are satisfied. Therefore, if we are to apply SGD to linear regression, we know that with probability one, $F(\theta_k) \to F_{\lim}$ which is finite. Since $F(\theta) \to \infty$ as $\theta \to \infty$, we know that $\{\theta_k\}$ cannot diverge. Hence, $\{\theta_k\}$ must remain finite with probability one. By Theorem 2, $\{\theta_k\}$ must converge to a stationary point. Since this stationary point is unique in our specific example of linear regression, we know that SGD must converge to the global minimizer of the linear regression problem. Note, we can follow this outline to draw similar conclusions in more complex situations.

## 5   Conclusion

In this work, we studied the global convergence analysis of Stochastic Gradient Descent with diminishing step size. We began our discussion by producing three simple problems for which the common assumptions (i.e., global Hölder continuity, bounded variance) and their generalizations in the SGD literature are violated. Indeed, to our knowledge, there does not exist theory that covers these problems. For example, prior to our work, it was unknown what SGD with arbitrary initialization and diminishing step sizes will do on simple neural network problems, which raised the question of what SGD is doing on more complicated learning problems.

Motivated by our example problems, we considered a more general set of assumptions (see Assumptions 2 and 4). Given the generality of our assumptions, we developed a new analysis technique that is of interest beyond this work, and we proved that SGD's iterates either converge to a stationary point or diverge. Thus, we now know how SGD with arbitrary initialization and diminishing step sizes will behave on a much larger class of learning problems.

We note that we do not provide rate of convergence results mainly because it is *impossible* for the broad class of functions admitted by our assumptions [Wolpert and Macready, 1997]. We stress that global rates of convergence (e.g., complexity statements) results that exist do not apply to the two simple neural network problems that we supplied at the beginning of this work.

We also studied what happens when SGD's iterates diverge. To this end, we required an additional assumption under which we developed another novel analysis technique and showed that, regardless of SGD's iterates' behavior, the objective function will converge to a finite random variable with probability one. Unfortunately, we make an assumption that we were not able to interpret, but we will leave this to future work.

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
