# OpenReview forum: "Global Convergence and Stability of Stochastic Gradient Descent"
_NeurIPS.cc/2022/Conference — NeurIPS 2022 Accept_

### Official Review · Reviewer_JXLB · 2022-07-06

**Rating:** 5
**Confidence:** 3
**Soundness:** 3 good
**Presentation:** 4 excellent
**Contribution:** 2 fair

**Summary:**

The authors analyze the convergence and stability of stochastic gradient descent under certain assumptions. In particular, they consider the function to be *locally* Holder-continuous (in opposition to global assumptions), and they relax the boundedness of the variance. In particular:

Under reasonable assumptions on the stochastic gradient descent (SGD) algorithm (with diminishing step size),  the authors show the iterates of SGD either converge (to a stationary point) or diverge to infinity. Under another technical assumption, they also show convergence with probability one.

**Questions:**

Do the authors believe it is possible to remove the assumption that max_k |theta_k| < infinity?

**Limitations:**

See the "strengths & weaknesses" section.

**Strengths And Weaknesses:**

The paper claims it uses less restrictive assumptions than other existing results. I agree with the authors; they generalize the analysis of SGD to a broader class of functions. However, my biggest concern is the (kind of hidden) assumption that most results hold under |theta_k|<infty.

Moreover, the result of Theorem 1 seems intuitive, given that we are working with diminishing stepsize. I may be wrong on this point, however. Here is my intuition.
Either:
the iterates are unbounded,
the iterates are bounded and, in that case, contained in a compact set of radius at most R. Since we have diminishing stepsize and a bounded variance, it seems reasonable to believe that, eventually, the iterates stop moving. Moreover, I think that in such a case, it is possible to write a rate of convergence for SGD to a stationary point that depends on R (which justifies Theorem 2).
In particular, the assumptions made in this paper are that f is locally Holder-smooth, and the variance is locally bounded - the authors justify introducing those hypotheses to contrast with similar results that assume global bounds. However, combining local assumptions and |theta_k|<R is (almost) the same as assuming global constants, as we can take the worst-case of those constants over R.

---

> ### Author Response · Authors · 2022-08-02
> **The intuition is mostly correct, the execution requires delicacy**
>
> The reviewer is *partly* correct about the intuition. We discuss the statements made by the reviewer below.
>
> # 1. Iterates are either bounded or diverge
>
> There are three cases, not just boundedness or divergence. We go through these cases first based on the reviewers comments before discussing the third.
>
> One cannot predetermine which $R$ we need to choose. Let us consider a one-dimensional example, $F(\theta) = 0$ and so $\dot F(\theta) = 0$, and we let $\dot f(\theta,X) = X$ where $X$ is a normal random variable. The iterates will converge to a normal random variable with mean zero and variance $\sum M_k^2$. Of course, there is no clear choice of $R$. This becomes a bit more interesting if we let $F(\theta) = \sin(\theta)$ and $\dot f(\theta,X) = \cos(\theta) + X$. Now, we can approach any of the elements in $\lbrace \frac{3\pi}{2}  + 2 k \pi : k \in \mathbb{Z} \rbrace$. Here, there is not apriori choice for $R$ that makes sense as we cannot know it. Thus, *even if the iterates are bounded, there does not exist an finite $R$ that guarantees all iterates remain within a ball of radius $R$ around the origin*. There is still another nuance that needs to be carefully handled discussed below in **5**.
>
> The second case is that the iterates can diverge. This actually offers several interesting possibilities, of which we provide sufficient conditions for "good" behavior through Assumption 5 and Theorem 3.
>
> The third case is that our iterates can visit infinity and return, and then go back. That is, the limit infimum and supremum are at the extremes. Think of a reflected random walk. With diminishing step sizes, this seems like it could not happen. But what if our noise is allowed to grow arbitrarily large as the iterates get farther and farther away? Then, the noise could outpace the diminishing step size. In effect, we could end up with something that behaves similar to a reflected random walk, in which the iterates get pushed back towards the origin (but maybe not too close) and then pushed towards infinity. *Because of the generality of the noise, especially asymptotically, this case is non-trivial to preclude, but is done through Theorem 1.*
>
> # 2. Diminishing Step-sized and Bounded Variance Implies Convergence
>
> The fact that we would converge to a stationary point should not be taken for granted. Consider the extreme of no noise and infinitely small step-size (i.e., continuous gradient descent). In this case, the iterates will converge to a limit cycle (see Palis-de Melo's book, Chapter 1, Example 3). The fact that stochastic gradient descent behaves strictly differently **was not apriori known**. For instance, the work of Mertikopolous et al. (2020) in NeurIPS does not say convergence to a stationary point as they could not discount the possibility of a limit cycle. We are able to preclude limit cycles and cycles in general. **Again, such a statement is not obvious, but we are able to prove it.**
>
> # 3. Choosing worst constants over all $R$
>
> Take the linear regression case. Here, the variance goes off to infinity as we must let $R$ go to infinity (see the previous point). So the worst case of the constant is $\infty$, and the argument would fall apart as you have stated it. One could artificially restrict the iterates to a region, and then say "with high probability", but this would be a weaker qualitative statement than what we have presented.
>
> # 4. Finding a Rate
>
> This would also be a qualitative statement as $R$ cannot be choosen apriori, so the rate can become arbitrarily bad. Moreover, we could try to make a "with high-probability" statement, but this would be a weak qualitative statement as we would not know where to set the cut-off or how awful $R$ would be. Thus, the rate can get arbitrarily bad as the constants can get arbitrarily bad (see **3** above).
>
> # 5. Arguments with bounded $R$
>
> Suppose we fix the set $R$ and we look at all iterate sequences that remain within $R$. How? This would require knowledge about the future of the iterates, so if we went to take (conditional) expectations, we would not even know how to calculate them.
>
> As we see, the intuition is partly correct and the natural arguments that flow from it fall apart quickly. Our work handles these subtleties carefully and thoughtfully, which we hope that we have conveyed. We hope the reviewer will reconsider their position.

---

> > ### Comment · Reviewer_JXLB · 2022-08-08
> > **Response**
> >
> > Dear authors,
> >
> > thank you for the detailed response and the clarification. I may have underestimated the difficulty of some technical details. I updated my score. This may change again during the discussion phase.
> >
> > Best regards,
> >
> > Reviewer JXLB

---

### Official Review · Reviewer_SUmp · 2022-07-09

**Rating:** 5
**Confidence:** 2
**Soundness:** 3 good
**Presentation:** 4 excellent
**Contribution:** 3 good

**Summary:**

This paper studies the convergence of stochastic gradient descent for smooth non-convex stochastic optimization. The main contribution of this paper is that it relaxes the commonly-used assumptions in proving the convergence: (1) global smoothness of the objective function, (2) bounded variance of the stochastic gradient. They are replaced by a much weaker assumption, i.e. local Hölder continuous of the gradient and bounded $1+\alpha$ moment of the stochastic gradient. The authors prove that SGD with a diminishing step size must either converge to a stationary point or diverge. They further add a technical assumption and prove a stability result, showing that the objective function cannot diverge even if the iterates diverge.

**Questions:**

N/A

**Strengths And Weaknesses:**

**Strengths**:
1. This paper is well-written. The motivation is clear and the presentation is concise.
2. It seems that the results and technical contribution are non-trivial.

**Weaknesses**:
The results seems to be not very interesting. While it gives convergence results, it does not say anything about the convergence speed (while the authors have pointed out that it is impossible under such general assumptions). In such a situation, my concern is that are the general assumptions really necessary? For example, modern optimization problems for neural networks are clearly not globally smooth, but relaxing it by locally Hölder continuous in gradients seems too general. Several works considered more restrictive assumptions, e.g. the generalized smoothness [1,2], under which optimal algorithms have been proved. Analogously, several works relax the assumption of bounded gradient variance by unbounded one that is locally proportional to the gradient magnitude [3,4] (which is satisfied by your first example), in which case optimal algorithms are also developed.

But this paper clearly has its own values. Currently I haven't delved into the proof details and I may not correctly understand the implications of these results, so I toned down the level of confidence.

[1] Why gradient clipping accelerates training: A theoretical justification for adaptivity. ICLR 2020.

[2] Improved Analysis of Clipping Algorithms for Non-convex Optimization. NeurIPS 2020.

[3] The Power of Adaptivity in SGD: Self-Tuning Step Sizes with Unbounded Gradients and Affine Variance. COLT 2022.

[4] Non-convex Distributionally Robust Optimization: Non-asymptotic Analysis. NeurIPS 2021.

---

> ### Author Response · Authors · 2022-08-02
> **More stringent assumptions do not always work via Examples**
>
> We thank the reviewer for providing references to support their points. We find that this is rarely done, and often claims are just made. So thank you for doing so. We address the comments below.
>
> For [1,2]: The ($L_0,L_1$)-Smooth condition is not satisfied by Example 2 (remove the penalty term for simplicity), which is a really simplified feed forward network. To see this, we can use any norms owing to the equivalence of norms. The Frobenius norm of the Hessian can be lower bounded by the absolute value of its [1,1] entry, which is
> $$\frac{0.5 W_4^2 W_3^2 W_2^2}{( 1+\exp(W_4 W_3 W_2 W_1) )^2}$$
>
> The $\ell^1$ norm of the gradient is
> $$\frac{0.5}{1+ \exp(W_4 W_3 W_2 W_1)} \left[ |W_4 W_3 W_2| + |W_4 W_3 W_1| + |W_4 W_2 W_1| + |W_3 W_2 W_1| \right]$$
>
> Now consider when $W_1 = -1$, $W_4 = W_3 = W_2$. Then, the lower bound on the norm of the Hessian is
> $$\frac{0.5 W_2^6}{ ( 1 + \exp(-W_2^3) )^2}$$
>
> And the gradient is
> $$\frac{0.5}{1 + \exp(-W_2^3)} \left[ |W_2^3| + 3|W_2^2|\right]$$
>
> Thus, as $W_2 \to \infty$, we see that the lower bound on the Hessian cannot be bounded by the gradient. Hence, this **simple** feed forward network does not satisfy the assumptions of [1,2], specifically the $(L_0, L_1)$-Smooth assumption. To suppose now that it would apply to more complicated, deeper feedforward networks would be hard to believe.
>
> For [3,4]. This assumption is generalized by expected smoothness from Peter Richtarik's group. Here is a simple example on a bounded domain where such a condition is not satisfied but our assumption allows us to optimize over the entire interval. Consider $f(\theta,X) = \theta^X$ where $X$ is an exponential random variable with parameter $1$ and let $\theta \in [1, e^{2/3}]$. Then,
> $$F(\theta) = \frac{1}{1 - \log(\theta)},$$
> which is minimized when $\theta = 1$. Moreover, its gradient is also computable and finite on the interval. Now, the variance of $\dot f$ does not exist on $[e^{1/2}, e^{2/3}]$, hence the conditions of [3,4] are not satisfied. However, our noise assumption **is* satisfied when we reduce the Holder constant to be less than 1/2. (Note, restricting SGD to the interval actually simplifies our analysis).
>
>
> We hope these examples shed light on why the analysis is needed, and why the referenced examples can fall short in some situations that are likely to occur (especially [1,2]).

---

> > ### Comment · Reviewer_SUmp · 2022-08-04
> > **Thank you for your response.**
> >
> > I would like to thank the authors for their detailed response. I do believe that these assumptions are still not satisfactory. My point is that the authors may strengthen the assumptions a bit which may yield a guarantee of convergence speed. Nevertheless, I have pointed out that this paper has its own value. So I will keep the rating unchanged.

---

> > > ### Author Response · Authors · 2022-08-04
> > > **Thank you**
> > >
> > > Thank you for your consideration.
> > >
> > > Just as a final point that is more important than the score: we hope that we have convinced the reviewer that the local Lipschitz assumption cannot be relaxed further in realistic ML contexts as shown by our examples (excepting when differentiability no longer exists, which would require **weaker** assumptions than local Lipschitz continuity of the gradient) and by our examples for the reviewer. As a result, owing to the "No Free Lunch Theorem" (Wolpert, D.H., Macready, W.G. (1997), IEEE Transactions on Evolutionary Computation), we cannot even have a rate of convergence even if the noise were to be zero (i.e., the deterministic context). Thus, any *realistic* rate of convergence would have to be very example dependent and would not apply to, say, a three-layer or deeper feed forward network that makes use of (smoothed) ReLU functions.

---

### Official Review · Reviewer_GcPb · 2022-07-11

**Rating:** 7
**Confidence:** 2
**Soundness:** 3 good
**Presentation:** 3 good
**Contribution:** 3 good

**Summary:**

The paper presents an analysis of stochastic gradient descent for a wide range of applications by removing several usual but problematic assumptions from the analysis, replacing them with more acceptable alternatives. The issues with the current assumptions are illustrated for 3 common problems. Somewhat surprisingly, the authors are able to prove convergence and stability under the new, more permissive assumptions, though not a convergence rate (which wouldn't be possible under these assumptions anyway).

**Questions:**

1. For 2 counter-examples, a Ridge penalty is mentioned. Is it strictly necessary? If that's the case, then this weakens the argument that previous assumptions were not good enough.
2. l92, where the authors claim that SGD basically always converges, seems somewhat surprising for deep learning practitioners. Can the authors shed some light as to why there seems to be a difference with practice (where SGD certainly can blow up)?

**Limitations:**

The authors mention the limitations of their work several times, notably making clear that the technical assumption that they require for their analysis cannot easily be interpreted. In that sense, I think limitations are honestly and properly addressed.

**Strengths And Weaknesses:**

The paper is well written and easy to follow (at least superficially). The improved assumptions mean that the analysis is a lot more widely applicable than previous attempts, which relied on violated assumptions and were thus of less value. The results are original and significant, as are the new methods used to prove the theoretical results.

I do have a few suggestions for improvements though:
- when presenting the 'counter-examples' in the main text, all the authors say is that they have a result that shows there is a counter-example, without even referring to the Appendix for the actual example. I wonder if a simplified version of the counter-examples themselves wouldn't improve the main text, which currently reads as 'trust me on this'.
- there is a lot of introductory space in the paper, and we only get to the meat of it on page ~6, with many results devolved to the Appendix. If the authors want a self-contained paper, then a journal format would be a better fit. For a conference paper, summarizing the first few pages more succinctly might read better.

---

> ### Author Response · Authors · 2022-08-02
> **Counter-examples and questions**
>
> Thank you for your comments!
>
> *Suggestion for Improvements*
> - On counter examples: line 36 says that the details are in Appendix A. We will can remind the reader of this again in the revision.
> - We will try to reduce the size of the introduction/sections 2/section 3 in the revision.
>
> *Questions*
> - No the ridge penalty is not required. We included it as this more closely approximates what is done in practice.
> - L92: There are two likely two things that can occur. First, Assumption 5 may not be satisfied (which we are working on, as we state in the limitations), which means the iterates may yet diverge. Second, there is a difference between the numerical behavior of SGD and its theoretical behavior. To be specific, Patel (2017) 'The impact of local geometry and batch size on the convergence and divergence of stochastic gradient descent' show that for large initial step sizes, SGD can diverge very far away from a minimizing region. If this happens, the numerical calculations can cause further problems before the theoretical convergence region of diminishing step sizes kicks in. Hence, even though we should expect SGD to converge *in theory*, the preliminary step sizes can cause SGD to first blow up and then the resulting numerical problems can prevent the convergence from occurring.

---

> > ### Comment · Reviewer_GcPb · 2022-08-08
> > **thanks!**
> >
> > thank you for your answer which mostly clears up the question I had. My opinion on the paper remains that it's a nice one which clears the bar for publication at NeurIPS this year.

---

### Official Review · Reviewer_3g83 · 2022-07-11

**Rating:** 5
**Confidence:** 3
**Soundness:** 3 good
**Presentation:** 2 fair
**Contribution:** 2 fair

**Summary:**

The paper considers the stability of SGD under more general conditions noise models than have been considered before. The paper first  demonstrates three common ML optimization were the convergence of SGD has not been established. Then the paper explores the convergence of SGD under quite general condition, e.g., relaxing both the assumption of Lipschitz continuous gradients and bounded variance, which are usually critical in establishing convergence results for SGD. The results establish that under such general conditions, SGD either convergence to a stationary point or they diverge to infinity. Finally, the paper suggests a technical condition that, if met, ensures convergence to a stationary point.

**Questions:**

I am a bit surprise that the SGD will always converge to a unique stationary point (given that it does not diverge). If there is a connected set of stationary points, is it not possible that the iterates do not converge to any one of them but rather move around around that set? I mean, e.g., if F:R—>R, if [0,1] is the set of stationary points with F’(\theta)=0 for all \theta in [0,1], I could imagine that the iterates of SGD would after some time start to move around randomly in [0,1]. According to your results, this is not possible? or I am I misunderstanding something?

**Limitations:**

No major limitations that I could see.

**Strengths And Weaknesses:**

Strengths: The paper is mostly well written. The paper is very upfront about the limitations of the work.  The mathematical results seem solid. The problem that is considered here is very general and it is indeed that it is possible to establish some convergence results in this case.

Weaknesses: It is difficult to see how the results could be useful in practice, because it is impossible to say before running the algorithm whether it converges to stationary point or diverges. In fact, the only thing the results in the paper exclude is that the iterate move around without diverging and without settling on a stationary point.

There are no-indepth discussion about how the insights here could be used in practice. There are also no experimental results.

---

> ### Author Response · Authors · 2022-08-02
> **Practicality of work; Questions**
>
> *Comment on perceived weaknesses:*
> As SGD is widely deployed, especially since it is one of two algorithms that achieves high quality generalization in deep, over-parametrized network (see Suvrit Sra's MIT Group's recent works), we critically need to understand what it is doing. We address your suggested weakness in two points.
> - There are examples that satisfy Assumptions 1 to 4 that will clearly let the iterates diverge and **for which this is the appropriate behavior**, for instance $\exp(-\theta^2)$. Hence, we cannot exclude it apriori. Perhaps more interestingly, there are examples for which the iterates will diverge and **for which it is not an appropriate behavior.** Thus, we have introduced Assumption 5 and its consequence (Theorem 3), which states that we will only converge to regions of zero gradient and finite objective function. Given that the three examples given satisfy Assumptions 1 through 5 and their objective function diverges as the argument goes to infinity, then we can state as a straightforward consequence of Theorem 3 that the iterates **cannot diverge**. Hence, we need only verify that the objective function satisfies Assumption 5  (say because of a regularization term in the objective, as noted in Lines 281 to 282) to ensure that the iterates do not diverge. This would certainly be a clear practical application, and we can expand on Lines 281 to 282 to make this clearer and to **add a deeper discussion about how to use our results in practice.**
> - There are a handful of possible behaviors: (1) divergence (ADAM can do this on a 1-dimensional convex problem), (2) convergence to a stationary point, (3) convergence to a nonstationary point (easy to construct examples), (4) entering a cycle (gradient descent can do this), (5) convergence to a limit cycle (continuous gradient descent can do this). Let's say that (3) or (4) could happen, then there would be no point in any saddle point or local maximum escape work, or even stopping conditions. Thus, we only want (1), (2) or (5) to be possible (only if 5 is a minimizing limit cycle). Apriori, *we do not know which outcome can happen* for SGD. To be able to say that only (1) and (2) can happen is somewhat surprising and useful for escaping non-miniming points and developing stopping conditions. The fact that (5) is precluded is also interesting as it goes against the intuition that continuous gradient descent can be used as a proxy for SGD (see works starting with Chizat & Bach 2018 that have not been as carefuly as this work).
>
> As this paper is primarily concerned with the theory of a widely deployed algorithm, we did not think it necessary to include experimental results.
>
> *Question:* I think the reviewer with this question points out the utility of this work in understand the behavior of SGD that we have provided. So in the example given, SGD is driven by IID mean zero noise with finite variance. Our theorem states SGD would then converge to a unique point in this region, which coincides with the conclusion of Kolmogorov's 3 series theorem.
>
> We hope that we have satisfied the reviewer's concerns and questions.

---

### Official Review · Reviewer_fVMv · 2022-07-15

**Rating:** 6
**Confidence:** 4
**Soundness:** 3 good
**Presentation:** 4 excellent
**Contribution:** 3 good

**Summary:**

This paper analyzes the global convergence of SGD under more practical assumptions, for the ease of studying the realistic performance for training with non-convex function and noisy feedback. Specifically: 1) it relaxes the global Lipschitz continuity of the gradient function to local $\alpha$-Holder continuity; 2) it relaxes the bounded variance assumption to that the $(1+\alpha)$-moment of the stochastic gradient is bounded by an arbitrary upper semi-continuous function. Hence, the authors provide a new analysis that SGD will either converge to a stationary point or diverge. To ensure the convergence, an additional assumption that generalizes the notion of expected smoothness is introduced.

**Questions:**

1. How does assumption 5 hold for the cases in Example 1,2,3?
2. It is better to give full definitions of $\alpha$-holder continuous and semi-continuous function.


**Ethics Review Area:**

["I don’t know"]

**Limitations:**

The main limitation is the interpretation of Assumption 5, which is left for future works.

**Strengths And Weaknesses:**

Strengths:

1. The authors provide three typical losses: linear regression, feed-forward neural networks and recurrent neural networks, to demonstrate the invalidation of current assumptions for analyzing the global convergence of SGD. This shows that current analyses can be applied to practical machine learning problems.
2. The authors give a solid analysis of the global convergence and high-level descriptions for the proof.
3. Theorem 1 is a strong result than I thought when I just read the introduction. Provided that the model parameters are usually finite, we can infer that the model converges globally from Theorem 1.

Weakness:

1. Proposition 1 is built by setting $\|\theta\|_2 \rightarrow \infty$, which is not the typical situation for training deep learning models, since the model parameters usually are finite during the training process.

---

> ### Author Response · Authors · 2022-08-02
> **Addressing comment on weakness and questions**
>
> For Proposition 1, the most common assumption is that the stochastic gradient variance is uniformly bounded for all $\theta$. The referenced survey by Patel [2021] provides references. As the example shows, this is clearly not true for linear regression because the variance goes to infinity as $\theta$ goes to infinity (i.e., the variance becomes unbounded), which implies that the uniform boundedness assumption on the stochastic gradient variance is not realistic.
>
> Assumption 5 holds for the given examples.
>
> We can add these definition to the rebuttal versions.
>
> We hope, given the pressing nature of understanding SGD in realistic scenarios in machine learning, and the addressing of the suggested weakness and questions, the reviewer is satisfied.

---

### Meta-Review · Area_Chair_mxqD · 2022-08-30

**Recommendation:** Accept
**Confidence:** Less certain

**Metareview:**

This paper analyzes the asymptotic convergence behavior of SGD on the class of locally Hölder continuous functions, by generalizing the technique and results of Patel [2021]. The paper extends and generalizes prior SGD analyses that were conducted under the assertion that certain conditions (e.g. smoothness or continuity) hold globally.

The reviewers found that the results are well presented, correct, and of interest to the community. The results could stipulate further research on SGD analyses on function classes more relevant to neural network training or deep learning, and also on non-asymptotic analyses of SGD on the function class studied in this paper.

The internal discussion brought up a few concerns should be carefully addressed when preparing the final version:
- some reviewers found the examples a bit overclaimed and not clearly showing the necessity of considering locally Hölder continuous functions. For instance, analyses that do not require a bounded variance assumption have become standard in recent years (see for instance the textbook by Bottou, Curtis, & Nocedal that is cited in the paper) and thus Example 1 (Linear Regression) could be a bit misleading as it is bringing up an already solved issue. Please relate the examples carefully to the (novel) contributions in this paper,
- and please mention and explain the relation to the arXiv preprint https://arxiv.org/abs/2104.00423.

**Award:**

No

---

### Decision · Program_Chairs · 2022-09-14

Accept